# The Pentameric Ligand-Gated Ion Channel Family: A New Member of the Voltage Gated Ion Channel Superfamily?

**DOI:** 10.3390/ijms25095005

**Published:** 2024-05-03

**Authors:** Aditi Dubey, Madison Baxter, Kevin J. Hendargo, Arturo Medrano-Soto, Milton H. Saier

**Affiliations:** Department of Molecular Biology, School of Biological Sciences, University of California San Diego, La Jolla, CA 92093-0116, USA; adubey@ucsd.edu (A.D.); mtbaxter@ucsd.edu (M.B.); khendarg@ucsd.edu (K.J.H.)

**Keywords:** VIC superfamily, pLIC family, TCDB, homology, bioinformatics

## Abstract

In this report we present seven lines of bioinformatic evidence supporting the conclusion that the Pentameric Ligand-gated Ion Channel (pLIC) Family is a member of the Voltage-gated Ion Channel (VIC) Superfamily. In our approach, we used the Transporter Classification Database (TCDB) as a reference and applied a series of bioinformatic methods to search for similarities between the pLIC family and members of the VIC superfamily. These include: (1) sequence similarity, (2) compatibility of topology and hydropathy profiles, (3) shared domains, (4) conserved motifs, (5) similarity of Hidden Markov Model profiles between families, (6) common 3D structural folds, and (7) clustering analysis of all families. Furthermore, sequence and structural comparisons as well as the identification of a 3-TMS repeat unit in the VIC superfamily suggests that the sixth transmembrane segment evolved into a re-entrant loop. This evidence suggests that the voltage-sensor domain and the channel domain have a common origin. The classification of the pLIC family within the VIC superfamily sheds light onto the topological origins of this family and its evolution, which will facilitate experimental verification and further research into this superfamily by the scientific community.

## 1. Introduction

All living cells rely on molecular transport to maintain key physiological processes and survive. Cellular transport depends on proteins located within membranes to form channels, carriers and pumps that regulate the passage of compounds and signals into and out of cells and organelles. Many transporters function by similar mechanisms, share topological features, and have common evolutionary origins, allowing them to be classified into families and superfamilies. These family classifications are tabulated and recorded in the Transporter Classification Database (TCDB; https://tcdb.org), which provides curated functional and evolutionary information on transporters across all domains of life and was adopted by the IUBMB over 20 years ago [1].

One important group of signal transduction membrane proteins is the Voltage-gated Ion Channel (VIC) Superfamily. Proteins in the VIC superfamily are found in all domains of life and are involved in the transport of select cations, including K^+^, Na^+^ and Ca^2+^ [2,3,4]. According to TCDB, the VIC superfamily currently consists of seven families (Table 1). The VIC family (TC: 1.A.1) mediates transport of K^+^, Na^+^, and Ca^2+^. Ion flux via the eukaryotic channels is generally controlled by the transmembrane electrical potential or by ligand or receptor binding [5,6,7]. Members of the VIC family have both homo-oligomeric and hetero-oligomeric structures with a variable number of subunits, but the α-subunit often comprises the channel and primary receptor. The α-subunits each have six TMSs (or multiples of these six TMS units, with 12 or 24 TMSs, each unit separated by a hydrophilic loop) [8,9,10]. VIC superfamily channels contain voltage sensors that regulate ion conduction [11]. These sensors consist of four TMSs [12] that include conserved charged residues, and TMS 4 includes conserved triplet repeats that each include one basic residue followed by two hydrophobic residues [13]. Voltage-gated Na^+^ and Ca^2+^ channels commonly have 24 transmembrane segments with a 4 × 6 TMS topology [13]. The K^+^ channels can be classified into four families in humans, which include (1) voltage-dependent K^+^ channels, (2) pore domain channels, (3) Ca^2+^-activated K^+^ channels, and (4) inward rectifier K^+^ channels [14,15,16,17]. The VIC family was the original family in the superfamily, but as more distant “subfamilies” were added, it proved to be part of a superfamily (Table 1) [18,19].

Members of the IRK-C family (TC: 1.A.2) have only the pore (P), lacking the voltage-sensing domain [22]. Each α-subunit has only two TMSs linked by a P-type re-entrant loop [23]. These channels differ from other K^+^ channels, as they allow K^+^ to flow more easily into the cell than out of the cell. One of the channels, Kir3, is related to heart rate regulation [24], while the G protein-gated inwardly rectifying K^+^ (GIRK) channel present in both the heart and brain binds phosphatidylinositol bisphosphate to remain in an open state [25].

The RIR-CaC family (TC: 1.A.3) members cause the release of Ca^2+^ from the endoplasmic reticulum (ER) in animal cells [26]. The Ryanodine (Ry) receptors (RyR) are usually found in the sarcoplasmic reticular membranes of muscle cells, while inositol 1,4,5-triphosphate (IP_3_) receptors are present in the ER of a larger set of cell types [27,28]. Both Ry and IP_3_ receptors are homo-tetrameric complexes with six TMSs in each channel domain, and these domains show basic structural similarities with other proteins in the VIC superfamily [29,30,31].

The TRP-CC family (TC: 1.A.4) includes cation channels that have been found in almost every tissue of the human body [32]. This family consists of seven subfamilies [33], all members having similar structures [34,35]. These eukaryotic-specific channels can be selective for Ca^2+^ or non-selective for inorganic cations [32]. They are important for brain development and synaptic transmission [32,36].

Members of the Trk family (TC: 2.A.38) are found in bacteria, yeast, and plants. These proteins have eight TMSs and are homologous to the channel domains in the VIC family, having this domain quadruplicated [37,38,39,40]. Some members of this family are also involved with the binding of NAD^+^ and NADH, which may affect gating [41]. TMS 8 may function in preventing conformational changes essential for channel activity while at the same time allowing secondary active transport [42].

The GIC family (TC: 1.A.10) contains homotetrameric or heterotetrameric complexes that mediate fast excitatory synaptic transmission in the (CNS) [43]. Proteins in the GIC family have a characteristic topology of 3 + 1 hydrophobic peaks, where the first three peaks are close to each other and the fourth is separated from the first three by a long hydrophilic region. The second peak is a reentrant loop that likely participates in channel formation and ion selectivity [43,44]. The TMS corresponding to the third hydrophobic peak has been found to play a role in coupling ligand binding to channel gating [45]. Glutamate-activated channels of the GIC family are classified into three main types, which are activated by (1) α-amino-3-hydroxy-5-methyl-4-isoxazole propionate (AMPA), (2) kainate, and (3) N-methyl-D-aspartate (NMDA) [46,47,48,49,50,51].

Members of the VPC family (TC: 1.A.51) are dimeric channels that have been characterized in mammals, ascidians, zebrafish, and frogs [52]. Proteins in the VPC family have TMSs 1–4, which function as both the voltage sensor as well as the ion channel [53] (see footnote to Table 1). Hv1 is an example of a voltage-gated proton channel that has the voltage-sensing domain but still expresses proton-selective ion channel activity [54]. Hv1 is important for sperm motility, apoptosis, and metastatic cancer [55,56].

Members of the pLIC family (TC: 1.A.9) are responsive to ligands including acetylcholine, serotonin, glycine, glutamate, and γ-aminobutyric acid (GABA). They are found in prokaryotes and animals [57,58]. Members of the pLIC family are homo- or hetero-pentameric, with the five subunits forming the pore-like structure [57,58]. Like GIC family members, pLIC channels have four hydrophobic peaks organized in a 3 + 1 topology, with no reentrant loops; all four peaks are transmembrane α-helices with the first three TMSs close to each other and the fourth TMS separated by a long hydrophilic domain. TMSs 1, 3 and 4 associate with each other while TMS 2 lines and forms the membrane pore. TMS 3 plays a role in determination of receptor assembly [59] and may control the speed and efficiency of channel gating [60,61], while TMS 4 plays a role in receptor activity and lipid sensing [62].

While the pLIC family had not previously been shown to be a member of the VIC superfamily, marginal sequence alignment scores and similar topology (3 + 1) for members of the pLIC and GIC families were preliminarily observed, and this study provides additional evidence supporting its membership within the VIC superfamily. We have applied multiple bioinformatics methodologies [63,64,65,66,67,68] to gain evidence of homology: (1) sequence similarity, (2) compatibility of TMS topology in the context of repeat units and overlap of hydropathy profiles, (3) shared domains, (4) conserved motifs, (5) similarity of HMM-profiles at the family level, (6) common 3D structural folds, and (7) protein tree analysis of the superfamily.

## 2. Results

### 2.1. Relationship between Families pLIC and VPC

We identified significant sequence similarity and topological differences between members of the pLIC and VPC families (Figure 1). The pLIC family has a 3 + 1 topology (Figure 1A,C), while the VPC family members have four TMSs that are not separated by a long hydrophilic region (Figure 1D,F). Interestingly, there are homologs in family pLIC that have five hydrophobic peaks in a 4 + 1 topology, although the fourth peak is not a TMS (Figure 1A,C). Smith–Waterman alignments between homologs of these two families produced significant scores (SSEARCH E-value: 8 × 10^−6^ and MPSAT GEV *p*-value: 1.01 × 10^−4^; Figure 1G; see Section 4.2), where the first three TMSs of the pLIC homolog align with the first three TMSs of the VPC homolog. We do not consider informative the alignment of the fourth hydrophobic peak because it is not a TMS in the pLIC family. Despite this observation, there is detectable similarity between HMM profiles of Multiple Sequence Alignments (MSAs) involving these regions (HHalign E-value: 9.4 × 10^−5^ and GEV *p*-value: 1.08 × 10^−4^; see Section 4.4). Note that multiple alignments only included the first three TMSs to remove the influence of the fourth hydrophobic peak in family pLIC and because the addition of the hydrophilic regions separating the last two TMSs in pLIC led to better HHalign E-values. The availability of multiple sequences generating reasonable similarity scores indicates that there are subpopulations within these families that preserve the signal of homology. In addition, the characteristic Pfam domain (PF02932) of the pLIC family was projected (see Section 4.3) with a marginal score (E-value: 2.5 × 10^−3^) to family VPC (Figure 1F), further supporting the relationship between these families.

In support of this relationship, the signature region (SR) identified between families pLIC, VPC and GIC (see Section 2.4 below) maps to the third TMS of family pLIC and partially covers the third and fourth TMSs of family VPC. This is explained, at least in part, by the observation that alignments between members of these families contain gaps in the regions covering TMSs 3 and 4 of VPC.

Although VPC does not have the long hydrophilic loop between TMSs 3 and 4, characteristic of the pLIC family (Figure 1), the transmembrane domains of these two families share the same structural fold (RMSD: 3.6 Å; TM-align: 0.55; coverage: 91.4%; Figure 2). Appendix A lists several alignments that support this observation (RMSD < 4 Å; TM-score > 0.5; coverage > 75%) and the hypothesis that families pLIC and VPC are homologous.

### 2.2. Relationship between Families pLIC and GIC

Families pLIC and GIC have a similar organization of their hydrophobic peaks, both having 3 + 1 topologies (Figure 3A,D), and both have a hydrophilic loop connecting the third and fourth peaks. Smith–Waterman alignments between homologs of these families (Figure 3C,F) generated significant scores (SSEARCH E-value: 1.6 × 10^−8^; MPSAT GEV *p*-value: 3.94 × 10^−7^; see Section 4.2) and cover all hydrophobic peaks (Figure 3G). Furthermore, the alignment of HMM profiles between these families generated a reasonable HHalign score (E-value: 1.3 × 10^−4^), which was supported by a GEV *p*-value of 6.3 × 10^−5^ (see Section 4.4). The characteristic Pfam domain (PF02932) of the pLIC family can be projected onto family GIC with marginal score (E-value: 7.9 × 10^−3^; Figure 3F; see Methods Section 4.3). This score can be explained, at least in part, by the poor alignment in the region of the second hydrophobic peak, which is a TMS in family pLIC, but a reentrant loop in family GIC. We identified a 35-residue signature sequence that maps to the third hydrophobic peak of both families pLIC and GIC (see Section 2.4 below). Interestingly, there are fusion proteins between families pLIC and GIC. For example, the N-terminal domain of GIC member P19490 (TC: 1.A.10.1.1) matches the full length of pLIC member P48167 (TC: 1.A.9.3.1), while the C-terminal domain characterizes family GIC. Altogether, this evidence suggests that these families are (1) homologous, (2) functionally related, and (3) may have coevolved.

Our analysis revealed a structural incompatibility between families pLIC and GIC, resulting in poor structural superpositions between these families. Our initial filters selected the top 184 high-scoring alignments (see Section 4.7). However, all but one of these alignments involved GIC-pLIC fusion proteins and were not informative for this study. We relaxed our filters using RMSD < 4.3 Å, TM-score > 0.42, and coverage > 50%, resulting in 208 additional alignments. After visual inspection, only 47 alignments did not involve fusions. However, in 40 of these alignments, the second α-helix in GIC was not solved in the structures. The second helix in multiple GIC members is noticeably less hydrophobic than its counterpart in the pLIC family. This is because the second α-helix in the GIC family is not transmembranal, but a reentrant loop (RL). Consequently, the orientations of the third and fourth α-helices in GIC are inverted relative to pLIC, generating poor scores and low-coverage alignments (Appendix A). Although the residues of the RL in GIC were resolved in six structures, they all have poor resolutions (4.9–6.7 Å). Taking this into consideration, we were unable to align the RL with the second transmembrane α-helix of pLIC structures. The structural incompatibility between these families can be observed in Appendix A, where pLIC member nicotinic acetylcholine receptor δ subunit (CHRND; 1OED) and GIC member Glutamate receptor 1 (Gria1; 6NJN) are superposed (RMSD: 4.06 Å, TM-align: 0.42547, and coverage: 61.42%). While the first helices of both structures align well, the second helix in GIC is an RL that disrupts the rest of the alignment because it does not align with any region in pLIC and inverts the orientation of helices 3 and 4. Note that the poor resolution of both structures (≥4 Å) may imply that an alignment between the second TMS in pLIC and the GIC reentrant loop might be possible when better-resolution GIC structures become available. However, the alignment between TMSs 3 and 4 will remain disrupted by the change of orientation. Although it is rare for homologous sequences to have different 3D structural folds, multiple cases have been documented in the literature. For example: (a) homologous domains of the LEKTI inhibitor show about 35% identity and have different structures [69]; (b) there are proteins that show global sequence similarities and dissimilar structures [70,71]; and (c) there are various examples of protein pairs in PDB showing sequence identity > 50% that produce 3D superpositions, with RMSD > 6 Å [72].

To this point, there are three lines of preliminary evidence suggesting that such structural incompatibility arose due to the evolution of an ancestral TMS into an RL. Firstly, Appendix A shows that the average length of the reentrant loops in the GIC family (18.61 ± 2.76 aa) is within one standard deviation of the average length of TMSs across the superfamily (21.05 ± 3.56 aa; see Section 4.7). If we use a much larger set of structures (412 RLs in 262 structures) and compare the mean lengths of TMSs and RLs within the same protein while correcting for biases in TmAlphaFold assignments, the mean difference in lengths is still within one standard deviation (Appendix A). Furthermore, a scatter plot of these data showed no significant correlation between the lengths of TMSs and RLs in the VIC superfamily (*r*^2^ = 0.1; Appendix A), indicating that although TMSs tend to be larger than RLs, it is not uncommon to find RLs larger than TMSs. Secondly, in some VIC families, the RLs can be as hydrophobic as the TMSs (e.g., multiple members of subfamily 1.A.1.16 in TCDB). Thirdly, even though many RLs in the VIC superfamily consists of a helix and a loop (Figure 4B and Appendix A), other configurations are found in the VIC superfamily. More specifically, some reentrant loops within the TRP-CC family have a Helix-Helix topology, where 2 small helices of similar lengths are separated by a 2-residue loop consisting mainly of Gly and Glu (Figure 4A; see Section 4.7). This suggests that a TMS can be turned into a reentrant loop by introducing a sharp turn in the middle of the TMS.

### 2.3. The VIC Repeat Unit

Our analysis identified a three-TMS repeat unit in a 3 + 1 + 3 topology within the VIC superfamily (see Section 4.8). Although the sixth hydrophobic peak is a reentrant loop, we observed multiple cases of three-TMS alignments between TMSs 1–3 and hydrophobic peaks 5–7. Figure 5 shows a representative example between homologs ion transporters A0A2S6H3L2 and A0A4P7XD37, where the alignment is significant (SSEARCH E-value: 1.9 × 10^−6^ and MPSAT GEV *p*-value: 4.76 × 10^−7^; see Section 4.2). The second TMS aligns with the sixth hydrophobic peak, which is congruent with our analysis of families GIC and pLIC (Figure 3). This alignment was not possible at the structural level for the reasons discussed above (Section 2.2). The three-TMS repeat unit is also supported by our analysis of families VPC, GIC, and pLIC. At the sequence and structural levels, the four TMSs in members of VPC align well with the first four TMSs of VIC members, whereas the first three TMS of VPC members align well with the first three TMS of pLIC members (Figure 1). Note that at the structural level, the influence of the large loop between TMSs 3 and 4 in family pLIC can be removed, and it is possible to obtain a four-TMS alignment between families VPC and pLIC (Figure 2). Lastly, the signature region identified (Section 2.4 below) maps to the third TMS in families VPC, GIC and pLIC, supporting that these TMSs are homologous. All in all, because the fourth TMS of pLIC aligns with the fourth TMS of family VPC (Figure 2) and GIC (Figure 3), it is likely that the ancestral repeat unit was 4 + 4 and VIC lost the eighth TMS. Thus, the three-TMS repeat unit and a 4 + 4 original topology constitute a fourth line of evidence suggesting that the second TMS in the second repeat unit evolved into a reentrant loop (RL). Interestingly, there is evidence, both at the sequence level and the structural level, indicating that the three-TMS repeat unit in the MFS secondary carriers may have derived from the channel domain of VIC superfamily members, with a subsequent interconversion of the original RL in VIC becoming a TMS in MFS [19,40,73].

### 2.4. Signature Region Identification within the VIC Superfamily

We identified a conserved 35-amino-acid-long motif or signature region (SR) using the MEME suite of programs (Figure 6; see Section 4.5). The SR consistently mapped to the third TMS of family pLIC (Figure 7A), partially covered the third and fourth TMS of family VPC (Figure 7B), and the third hydrophobic peak (second TMS) of family GIC (Figure 7C). This finding further supports an ancestral 4 + 4 repeat unit in the VIC superfamily, where family VPC matches the first four TMSs, and families GIC and pLIC match the last four TMSs. The similarity between both repeat units is described in Section 2.1, Section 2.2 and Section 2.3.

The recovery rates for this SR in homologs of families GIC and VPC were 80.77% and 79.1%, respectively (MAST E-value ≤ 10^−4^). In contrast, the recovery rate in family pLIC was significantly lower (38.15%), suggesting that some of the subfamilies are considerably distant. This is supported by previous reports referring to this family as a superfamily [74]. This observation is explored in greater detail in Section 2.5 (The VIC superfamily tree) below. The data used to calculate the recovery rates of the SR are presented in Appendix A.

### 2.5. The VIC Superfamily Tree

To investigate the evolutionary relationships inside the VIC superfamily, a phylogeny including family pLIC was built using MrBayes [75,76] (see Section 4.6). Unfortunately, the distant relationships among some of these families prevented the generation of a reliable MSA, and our phylogenetic reconstruction did not converge (standard deviation of split frequencies > 0.01) after 2 million generations. This is not unexpected in a superfamily analysis, given that phylogenies rely on MSAs, and the more distant the sequences in a phylogenetic reconstruction, the less reliable the MSAs [77,78]. For this reason, we performed clustering analysis of pairwise Smith–Waterman bit scores using the program mkProteinClusters [64] (see Section 4.6). We have previously shown a remarkable overall topological similarity at the major branches between this method and phylogenetic reconstructions [64]. The radial tree in Appendix A illustrates the relationships between the transmembrane domains of all 458 established members of the VIC superfamily and the pLIC family in TCDB (agglomerative coefficient: 0.995). Family pLIC has its own major branch, and families GIC and VPC are the most distant families of their respective branches. Although this tree does not show an obvious relationship between families pLIC, GIC and VPC, it is evident that the closest families to pLIC’s branch are GIC and VPC. This explains why the best homology signals for the pLIC family were produced by the GIC and VPC families. We hypothesize that GIC and VPC could not cluster with pLIC because of the sequence divergence observed within family pLIC. That is, the Ward agglomerative method added families GIC and VPC as the last groups of their respective branches, because the resulting variance was smaller than if they were added to the pLIC branch. Note in Appendix A how the pLIC clade is composed of two densely populated subbranches.

To test this hypothesis, we identified a subpopulation of 25 proteins within each of the pLIC, GIC and VPC families that better illustrate their relationship (see Section 4.6). The resulting tree (agglomerative coefficient: 0.993) is shown in Figure 8. In this case, a more obvious relationship between pLIC, GIC, and VPC can be seen, as these three families share the same branch. This confirms the existence of subpopulations within these families that preserve better the signal of homology. That is, a subpopulation of proteins that passed the different criteria used in this study to support homology. As homologous families diverge and become more distant, the subpopulation of proteins that preserve the footprint of their common ancestry is expected to become smaller. Thus, the problem of identifying remote homology is to track subpopulations of proteins preserving various features that reliably link diverging families.

## 3. Discussion

The evidence presented in this report supports our hypothesis that the pLIC family is a member of the VIC superfamily. We identified multiple lines of evidence supporting this conclusion. First, individual homologs were identified that show significant sequence alignments in the transmembrane domains (E-values < 10^−5^) between family pLIC (TC: 1.A.9) and families VPC (TC: 1.A.51) and GIC (TC: 1.A.10) (Figure 1 and Figure 3). Second, Pfam domains could be projected between families with marginal scores (Figure 1 and Figure 3). Third, we identified a three-TMS repeat unit in a 3 + 1 + 3 topology within the VIC superfamily (Figure 5) that presumably resulted from an ancestral 4 + 4 topology, followed by loss of the eighth TMS in some families. Fourth, a signature region (Figure 6) was identified (E-value < 10^−800^) that consistently mapped to the third TMS across families (Figure 7). Fifth, 3D structural analysis shows that the transmembrane regions of families pLIC and VPC (Figure 2; RMSD < 4.0 Å, TM-align > 0.5) share the same fold and suggested that the reentrant loop in the VIC superfamily evolved from an ancestral TMS (Figure 4A). Finally, cluster analysis of the superfamily (agglomerative coefficient: 0.993; Figure 8) highlights the existence of subpopulations within pLIC, GIC, and VPC that preserve the signal or footprint of their common ancestry.

Unfortunately, reliable 3D structural alignments were not obtained between families pLIC and GIC. This was due to poor-resolution GIC structures (>4.9 Å) with the reentrant loop solved and the structural incompatibility caused by the reentrant loop (RL) in GIC structures inverting the orientation of the last two TMSs (Appendix A). As more GIC structures with higher resolutions that include the RL become available, it will be possible to determine if the second TMS in pLIC can be aligned to the RL in GIC. We suggest, based on four lines of preliminary evidence, that the RL in GIC evolved from the second TMSs in the VIC superfamily: (a) similar lengths of TMSs and RLs (Appendix A), (b) the fact that RLs can be as hydrophobic as TMSs, (c) the identification of RLs with a Helix–Helix configuration (Figure 4A), and (d) the identification of an original four-TMS repeat unit organized in a 4 + 4 topology suggests that the second TMS in the first four TMSs is homologous to the RL in the second repeat unit (Figure 5 and Section 2.3). While two TMSs (or one loop) transforming into a reentrant loop would not change the orientation of the remaining TMSs relative to the membrane plane, the transformation of one TMS into a reentrant loop, has more profound structural consequences that need to be further studied in bioinformatic and experimental settings. The possibility of TMS-reentrant loop evolution has been proposed previously for families unrelated to this study [40].

The existence of fusion proteins between families pLIC and GIC suggests that, besides being homologous, members of these families are functionally related and may have co-evolved. Future work in this direction may investigate whether proteins from these two families are also coregulated. For example, if they can be found in the same operon, that would confirm their functional association. Alignments involving fusion proteins were excluded from this study because they could generate false positive signals.

The phylogenetic analysis of the VIC superfamily and the pLIC family did not produce a reliable tree because MrBayes did not converge (see Section 4.6). This is likely due to the distant relationships among families preventing the construction of adequate multiple alignments [77,78]. However, our cluster analysis showed that despite substantial sequence diversity within the pLIC family (Appendix A), there are subpopulations of homologs within families pLIC, GIC, and VPC that preserve the footprint of their common origin (Figure 8). The level of sequence diversity observed in the pLIC family explains why the recovery rate of the signature region in pLIC homologs (~40%) was significantly lower compared to families GIC and VPC (~80%) (Figure 6 and Appendix A). This could be due to the presence of distant subfamilies, functional differences or the implication that the pLIC family may itself be a superfamily [74].

The signature region (Figure 6) consistently maps to the third hydrophobic peak of families pLIC, GIC, and VPC while matching no other family in TCDB outside the VIC superfamily (Figure 7; see Section 4.5). This indicates that this sequence is confined to the VIC superfamily. TMS 3 in the pLIC family proteins is known to play a role in determination of receptor assembly [59] and has been found to undergo a conformational change that contributes to channel gating [61]. TMS 3 in GIC proteins is known to act as a transduction element, undergoing a conformational change to couple ligand binding to channel opening [45,79]. The role of TMS 3 in the VPC family proteins is less well understood; however, it has been proposed that TMS 3 is involved in voltage-sensing and signals for conformational change upon membrane depolarization [80]. Since the third TMS plays a role in channel gating in families pLIC, GIC, and VPC, the identification of the signature region covering the third hydrophobic peak is significant.

We are confident that the several lines of evidence presented in this study support the membership of the pLIC family within the VIC superfamily, setting the basis for experimental verification and further research into the evolution and functional divergence of this superfamily.

## 4. Methods

### 4.1. Sequence and Structural Data Retrieval

Families and their sequences were extracted from TCDB [1]. Homologs for each family were retrieved from the NCBI non-redundant database [81] and UniProt [82] with our program famXpander [64], using a blast E-value < 10^−4^ and a minimal coverage of 60% of the shorter sequence. Redundant sequences were removed with CD-HIT 4.8.1 [83] with an identity threshold of 90%. Protein domains were retrieved from Pfam [84]. Structural data were extracted from PDB [85], AlphaFold [86] and tmAlphaFold [87].

### 4.2. Sequence Similarity between Families

Our approach relies on the transitivity property of homology. That is, if protein A is homologous to protein B, B is homologous to protein C, and C is homologous to protein D, then proteins A and D are homologous even if no apparent similarity can be observed directly between protein A and D. Given that sequence similarity alone is not sufficient to reliably infer homology between distantly related transporter proteins [88,89], our strategy takes into consideration multiple lines of evidence [63,64,65,66,67,68]: (1) a path of sequence similarity across the transitivity path A → B → C→ D; (2) compatibility of TMS topology in the context of repeat units and overlap of hydropathy profiles; (3) shared domains; (4) conserved motifs; (5) similarity of HMM-based sequence profiles at the family level; (6) common 3D structural folds, and (7) protein tree analysis of the superfamily.

We compared sets of homologs between pairs of families with the program Protocol2 [90], which aligns sequences using the Smith–Waterman algorithm as implemented in SSEARCH 36.3.8h [91]. Top scoring Protocol2 matches were selected if the alignment covered at least three TMSs of the channel domain in the pLIC family. TMSs were predicted with HMMTOP 2.1 [92]. Overlap of hydropathy profiles in transmembrane domains was verified with the program QUOD [66].

When comparing transporter proteins, standard sequence alignment algorithms may yield scores beyond significance thresholds between unrelated proteins due to inherent compositional biases toward hydrophobic residues in transmembrane regions [88,89]. These biases are due to physico-chemical constraints within the membrane environment. To evaluate the significance of the observed Smith–Waterman scores, we used the inhouse program Membrane Protein Sequence Alignment Tool (MPSAT) [68], which is designed to account for compositional biases. MPSAT compares the alignment score between a query and subject proteins against a negative control of non-homologous, strategically shuffled versions of the subject proteins that preserve the amino acid composition, topology and distances between TMSs of the unshuffled subject protein. MPSAT then calculates a *p*-value from the Generalized Extreme Value (GEV) distribution of alignment scores between the query protein and the shuffled sequences to quantify the probability that the observed score between the biological (unshuffled) query/subject proteins could have arisen by chance [68]. This analysis helped us determine if similar topologies and localized compositional biases in TMSs account for the scores observed in our alignments. The MPSAT approach to calculate GEV *p*-value is generalizable to the comparison of multiple sequence alignments, as described in Section 4.4 below.

### 4.3. Projection of Pfam Domains

We used the program getDomainTopology [66] to characterize and project Pfam domains between families. Domain projections were used when sequences did not produce direct hits with a Pfam domain using the program HMMSCAN 3.4 and the gathering threshold [93]. The program getDomainTopology performs two iterations of sequence comparisons to project domains [66]. We only considered domain projections that succeeded in the first iteration.

### 4.4. Comparison of HMM Profiles of Multiple Alignments

Homologs generating top sequence alignments between pairs of families (Section 4.2) that passed our filters (E-value < 10^−4^, hydropathy overlap of TMSs and domain projections) were used as seeds to retrieve homologous sequences of the aligned regions with famXpander (E-value ≤ 10^−15^ and coverage > 95%), and multiple sequence alignments (MSAs) were constructed with the program MAFFT 7.490 using the L-INS-i algorithm [94]. The program Seqotron 1.0.1 [95] was used to edit MSAs and insert the gaps generated by the Smith–Waterman algorithm as implemented in SSEARCH 36.3.8h [91] for the two seed sequences. This is because the program HHalign from the HH-suite 3.3.0 [96] may not find the optimal alignment when comparing HMM models of distant families. HHalign compares HMM profiles generated from multiple sequence alignments between pairs of families. An alignment was regarded as significant when the E-value was ≤10^−4^.

We tested the reliability of the HHalign scores using the same rationale underlying MPSAT [68]. That is, based on the MSAs obtained based on both seed sequences, we generated 10,000 shuffled MSAs of the subject MSA such that the TMS topology, composition and distances between TMSs in the unshuffled subject MSA were preserved in all shuffled MSAs. Then, HHalign was used to generate alignment scores between the query MSA and all of the shuffled versions of the subject MSA. Lastly, we used the generalized extreme value distribution, as it fits well the distribution of alignment scores [97,98], to compute the probability (GEV *p*-value) that the HHalign score of the unshuffled alignments lies within the distribution of shuffled alignment scores, that is, the probability that a score in the fitted (analytic) distribution would be greater than or equal to the score of the unshuffled MSA. The Python library SciPy was used to fit a generalized extreme value distribution to the scores of shuffled proteins and calculate the GEV *p*-value, as described by Hendargo et al. [68].

### 4.5. Motif Analysis

The MEME suite 5.1.0 [99] was used to search motifs that were present across the VIC superfamily and family pLIC. Two member families of the VIC superfamily were selected (e.g., GIC and VPC) based on the quality of single- and multiple-sequence alignments, topological similarities, and domain projections (Section 2.1 and Section 2.2). We took a random sample of 100 homologs of typical length (excluding fusion proteins) from each of these three families and ran MEME to identify significant motifs of 8 to 50 amino acyl residues long (E-value ≤ 10^−700^) where each sequence contributed one motif (OOPS mode). We selected motifs whose positions mapped inside the channel domain and scanned them with MAST (E-value ≤ 10^−4^) on two datasets consisting of (a) all homologs from the three families (test set; 24,767 proteins) and (b) all other proteins in TCDB excluding the VIC superfamily (negative control). We found one motif (Figure 6) that mapped consistently to the third TMS in families pLIC, VIC and VPC while producing no matches against the negative control (Figure 7).

### 4.6. Phylogeny and Clustering Analysis

To better understand the relationship between the pLIC family and preexisting members of the VIC superfamily, we first attempted to build a phylogenetic tree. All members of each family were extracted from TCDB to generate a set of 565 proteins. We filtered these to keep only 458 proteins that possessed transmembrane domains. Then, the transmembrane region of each sequence was extracted using TMweaver [68], which allows the selection of region(s) in a sequence or MSA, via a graphical user interface, and returns either the coordinates or sequences of the selected region(s). To generate the phylogeny, an MSA of transmembrane regions was constructed with all families using the L-INS-i algorithm as implemented in MAFFT 7.490 [94]. Poorly aligned positions with more than 30% gaps were removed using trimAL 1.4 [100]. Finally, MrBayes 3.2.7a [75,76] was used, assuming different substitution rates per position that followed a gamma distribution with four rate categories. Posterior probabilities were estimated using Metropolis coupling (one cold and three heated chains) with 2 million generations. To test for convergence, the standard deviation of split frequencies needed to fall below 0.01. Unfortunately, the phylogeny did not converge due to the significant amount of sequence divergence observed across families, which not surprisingly produced a low-reliability MSA [77,78]. This problem was further exacerbated by the fact that some families lack the first VIC repeat unit (e.g., GIC and LIC) while others lack the second (e.g., VPC).

As an alternative approach, we performed a protein clustering analysis with the program mkProteinClusters [64]. This program uses the statistical computing environment R 4.3.1 (https://www.R-project.org/; accessed on 18 March 2024) to perform hierarchical clustering using the Ward method, based on a distance matrix calculated from bit scores generated by pairwise Smith–Waterman alignments as implemented in SSEARCH 36.3.8h [91]. This method has shown excellent topological agreement, at the major branch level, with phylogenies [64]. We created two trees: (a) using all 458 membrane regions from the pLIC and VIC families (Appendix A), and (b) substituting the transmembrane regions of families pLIC, GIC, and VPC with 25 sequences obtained from homologs of families pLIC, GIC, and VPC that generate significant signals of homology among these families (see Methods Section 4.4 above; Figure 8). The resulting trees were then analyzed using FigTree 1.4.4 (http://tree.bio.ed.ac.uk/software/figtree/; accessed on 1 April 2024).

### 4.7. Structural Analyses

We extracted from PDB [85] 812 protein structures of the pLIC family and VIC superfamily. Then, we used the program Deuterocol [1] to split them by chain and superpose them. This generated a total of 2154 chains. Structures were aligned with the TM-align program [101]. A total of 3,891,035 alignments were generated with TM-align [101] between protein chains of the pLIC family and members of the VIC superfamily. Alignments were ranked based on TM-scores and coverage. The top scoring alignments were visualized using PyMOL 2.5.0 and then inspected manually to ensure proper alignment and orientation of the transmembrane α-helices.

To investigate the potential evolution of a TMS into a reentrant loop, we used famXpander [64] to run DIAMOND 2.1.8 [102] against AlphaFold for proteins belonging to the VIC superfamily and the pLIC family. We selected proteins with E-value < 1 × 10^−7^ and a minimum query coverage of 50%, and redundant sequences were removed using an identity threshold of 90%. Hits with transmembrane segment/reentrant loop information registered on TmAlphaFold [87] were retained. Proteins failing to pass all of TmAlphaFold’s quality tests [87] were discarded from further analysis. The remaining predicted structures were subjected to an all vs. all structural alignment using TM-align v20160521 [101]. The resulting TM-scores were used to compute a distance (*d*) between pairs of proteins, *p*_1_ and *p*_2_, using Equation (1). A tree was then generated with the Scikit-learn library [103] using the average method and a TM-score threshold of 0.8 (equivalent to *d* ≤ 0.04).
(1)$$d\left(p_{1},p_{2}\right)={\left(1-TMscore\right)}^{2}$$

Pymol (https://pymol.org) was then used to inspect the resulting clusters and refine their TMS/reentrant loop assignments. Up to 30 structures were selected from each cluster and loaded into Pymol with raw TmAlphaFold assignments and suggested adjustments based on proximity. Additional residues were added to reentrant loops and TMSs if their alpha carbons were both within 1.5 Ångströms and within three residues of alpha carbons in existing reentrant loops or TMSs (Appendix A). Means and standard deviations for TMS and reentrant loop counts and lengths were then computed from manually corrected assignments (Appendix A).

### 4.8. Identification of the VIC Repeat Unit

We extracted 5,146 non-redundant homologs of family VIC (TC: 1.A.1) from UniProt using famXpander (E-value < 10^−4^ and minimal coverage of 80%). Redundant sequences were removed with CD-HIT 4.8.1 [83] using an identity threshold of 90%. Only homologs of VIC members with seven hydrophobic peaks were considered. We took three random samples without replacement of 500 sequences to generate multiple sequence alignments (MSAs) using the L-INS-i algorithm as implemented in MAFFT 7.490 [94]. MSAs were then trimmed with trimAL 1.4 [100] to keep positions with less than 30% gaps. To search for repeats we applied AncientRep [90] as previously reported [66]. AncientRep cuts each MSA in two MSAs, A and B, based on a given residue position, in this case we used the loop after TMS 4 as the cutting point. Then, for each reference MSA, all sequences in MSA A were individually aligned to all sequences in MSA B using the Smith-Waterman algorithm as implemented in SSEARCH 36.3.8h [91]. Only alignments with SSEARCH E-value < 10^−3^ were further considered. We identified 26 high-coverage alignments where the first 3 TMS in MSA A fully aligned with the 3 TMSs in MSA B. A representative example is shown in Figure 5. The alignment scores were verified with MPSAT [68] as described in Methods Section 4.2 above.

## Figures and Tables

**Figure 1 ijms-25-05005-f001:**
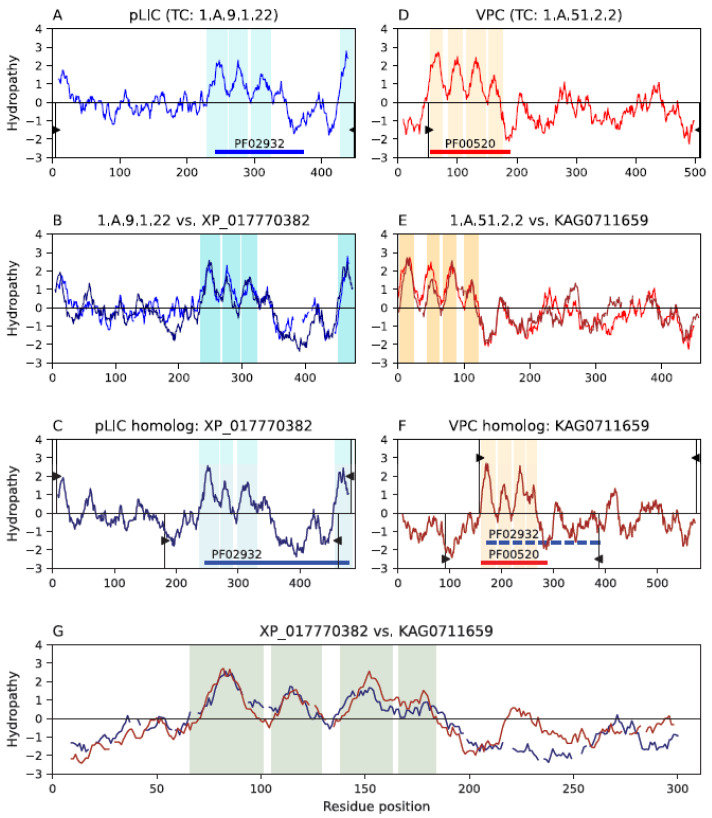
Homology between families pLIC and VPC. Hydropathy plots are presented across the homology transitivity path between families pLIC (TC: 1.A.9) and VPC (TC: 1.A.51), the latter being an established member of the VIC superfamily. Panels (**A**–**C**) depict relationships within family pLIC. Panels (**D**–**F**) depict relationships within family VPC, and panel (**G**) presents the evidence supporting homology between both families. Cyan and orange bars denote hydrophobic peaks (i.e., inferred TMSs). Only the regions where hydrophobic peaks overlap are highlighted in the alignments. Pfam domains are shown as colored horizontal solid or dashed bars. Thin vertical black lines with wedges delimit the regions of a protein involved in alignments. The wedges in panels (**A**,**D**) delimit the regions covered by the alignments in panels (**B**,**E**) relative to the full-length proteins in panels (**A**,**D**), respectively. Proteins in panels (**C**,**F**) have two sets of delimiting wedges. Wedges plotted for positive hydropathy values delimit regions covered by the alignments in panel (**B**) or (**E**) relative to the full-length proteins in panels (**C**,**F**), respectively. Wedges plotted for negative hydropathy values delimit regions covered by the alignment in panel G relative to the full proteins in panels (**C**,**F**), respectively. Interruptions in the hydropathy curves of panels (**B**,**E**,**G**) indicate gaps in the corresponding sequence alignments. (**A**) Hydropathy plot of pLIC member neuronal acetylcholine receptor α10 subunit (a10nAChR; Q9GZZ6; TC: 1.A.9.1.22). (**B**) Hydropathy plot of the alignment (E-value: 1.3 × 10^−72^) between pLIC member 1.A.9.1.22 and its homolog XP_017770382. (**C**) Hydropathy plot of pLIC homolog XP_017770382 extracted from NCBI. (**D**) Hydropathy plot of VPC member phosphatidylinositol 3,4,5-trisphosphate 3-phosphatase (TPTE2; XP_018103475; TC: 1.A.51.2.2). (**E**) Hydropathy plot of the alignment (E-value: 1.2 × 10^−83^) between VPC member 1.A.51.2.2 and its homolog KAG0711659. (**F**) Hydropathy plot of VPC homolog KAG0711659 extracted from NCBI. (**G**) Hydropathy plot of the alignment (E-value: 2.8 × 10^−6^; MPSAT GEV *p*-value: 1.01 × 10^−4^) between pLIC homolog XP_017770382 and VPC homolog KAG0711659. Alignments in panels (**B**,**E**) between the TCDB proteins and their NCBI homologs for pLIC and VPC, respectively, show that each homolog is clearly a member of its family. The alignment in panel (**G**) between the homologs in panels (**C**,**F**) presents the evidence supporting homology between both families. The Pfam domain drawn in dashed blue bars represents the projection with marginal score (E-value: 2.5 × 10^−3^) of the pLIC domain onto the VPC family.

**Figure 2 ijms-25-05005-f002:**
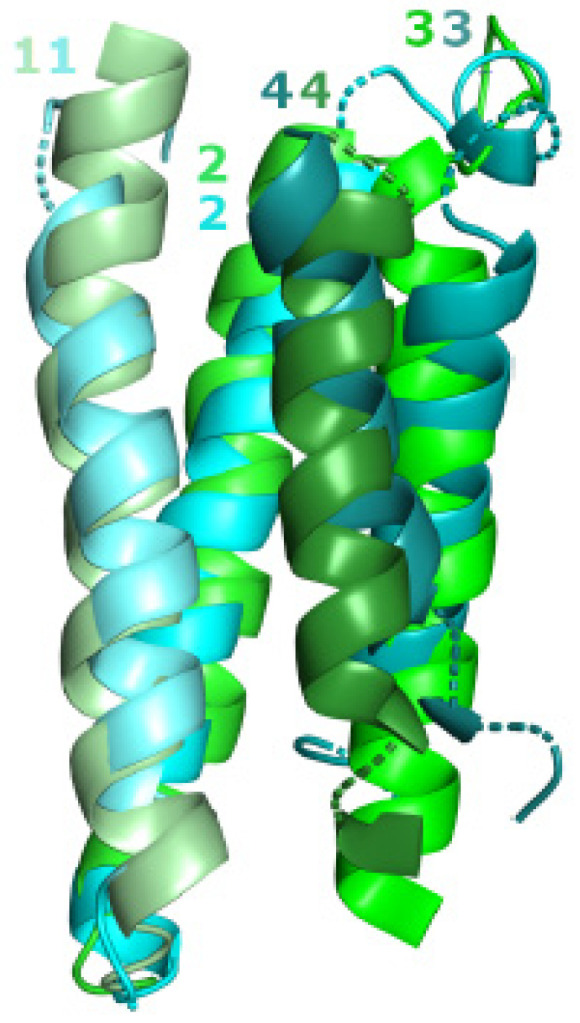
Common structural fold between families pLIC and VPC. Three-dimensional structural superposition of transmembrane regions between pLIC member nicotinic acetylcholine receptor α7 subunit (a7nAChR; 2MAW; cyan) and VPC member voltage-sensor containing phosphatase (Ci-VSP; 4G7V; green). Transmembrane α-helices 1–4 for 2MAW, chain A, and 4G7V, chain S, are labeled and shaded from light to dark in the order they appear in the amino acid sequence. For clarity, the long hydrophilic loop between TMSs 3 and 4 in the pLIC member was removed. The scores for the alignment are RMSD: 3.66 Å, TM-align: 0.55317, and coverage: 91.40%.

**Figure 3 ijms-25-05005-f003:**
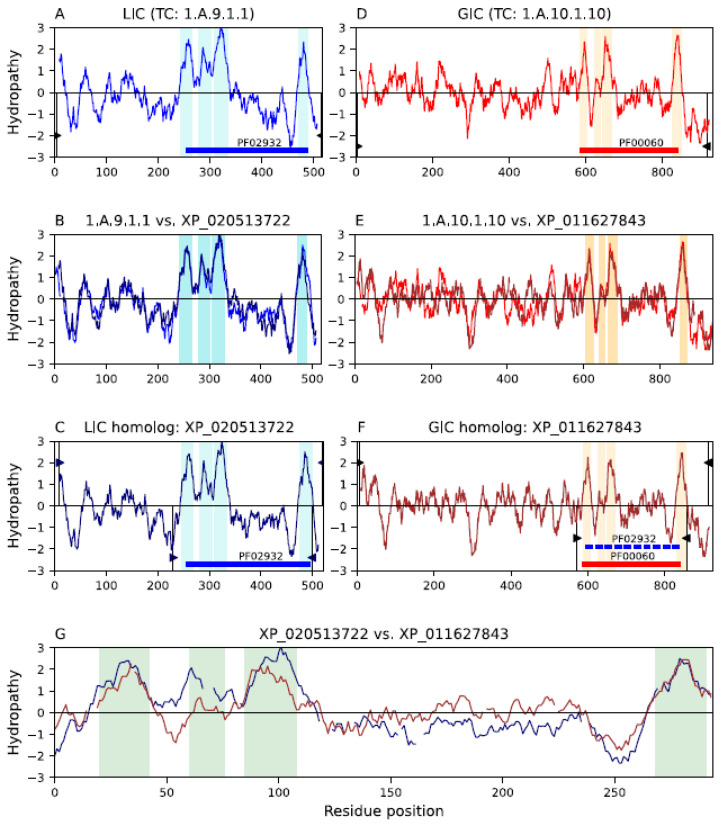
Homology between families pLIC and GIC. Hydropathy plots are presented across the homology transitivity path between families pLIC (TC: 1.A.9) and GIC (TC: 1.A.10), the latter being an established member of the VIC superfamily. This figure follows the format as described in the legend for Figure 1. (**A**) Hydropathy plot of pLIC member acetylcholine receptor δ subunit (CHRND; Q07001; TC: 1.A.9.1.1). (**B**) Hydropathy plot of the alignment (E-value: 1.2 × 10^−148^) between pLIC member 1.A.9.1.1 and its homolog XP_020513722. (**C**) Hydropathy plot of pLIC homolog XP_020513722 extracted from NCBI. (**D**) Hydropathy plot of GIC member glutamate receptor 3.3 (GLR3.3; Q9C8E7; TC: 1.A.10.1.10). (**E**) Hydropathy plot of the alignment (E-value: 1.6 × 10^−113^) between GIC member 1.A.10.1.10 and its homolog XP_011627843. (**F**) Hydropathy plot of GIC homolog XP_011627843 extracted from NCBI. (**G**) Hydropathy plot of the alignment (E-value: 1.6 × 10^−8^; MPSAT GEV *p*-value: 3.94 × 10^−7^) between pLIC homolog XP_020513722 and GIC homolog XP_011627843. Alignments in panels B and E between the TCDB proteins and their NCBI homologs for pLIC and GIC, respectively, show that each homolog is clearly a member of its family. The alignment in panel (**G**) between the homologs in panels (**C**,**F**) presents the evidence supporting homology between these two families. The Pfam domain drawn in dashed blue bars represents the projection (E-value: 7.9 × 10^−3^) of the pLIC domain PF02932 onto family GIC.

**Figure 4 ijms-25-05005-f004:**
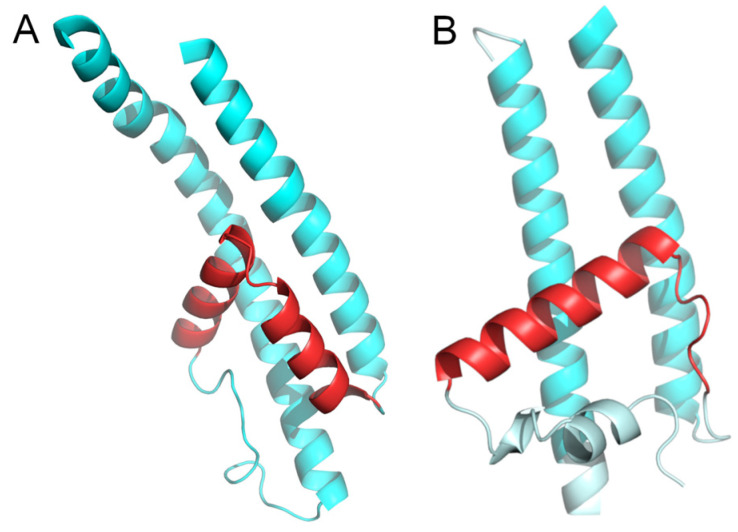
Configurations of reentrant loops in the VIC superfamily. Reentrant loops are presented in red and adjoining TMSs in cyan. Both structures are high-confidence AlphaFold inferences (pLDDT > 90) in the regions shown. (**A**) Helix–helix configuration of the reentrant loop in representative TRP-CC (TC: 1.A.4) homolog ankyrin repeat-containing domain protein (A0A8R1Y174). (**B**) Helix–Loop configuration of the reentrant loop in representative VIC (TC: 1.A.1) homolog: cyclic nucleotide-gated ion channel (I1K949).

**Figure 5 ijms-25-05005-f005:**
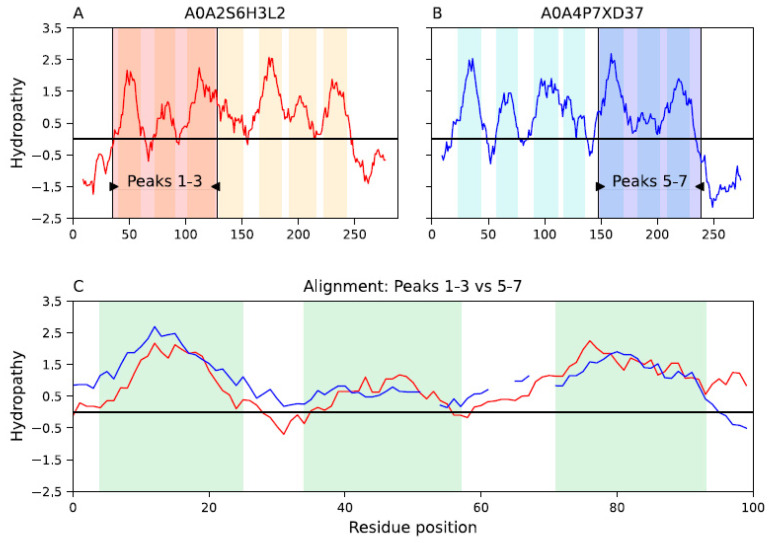
The VIC repeat unit. Representative alignment between two VIC ion transporter homologs supporting a basic three-TMS repeat unit. Hydrophobic peaks (i.e., TMSs or reentrant loops) are colored as vertical tan, cyan or green bars. Vertical black lines with wedges in panels (**A**,**B**) delimit the regions, colored in red or blue, that were aligned between homologous proteins. (**A**). The first three TMSs of the VIC homolog A0A2S6H3L2. (**B**). The last three hydrophobic peaks of VIC homolog A0A4P7XD37. The sixth peak is a reentrant loop. (**C**). The alignment (SSEARCH E-value: 1.9 × 10^−6^; MPSAT GEV *p*-value: 4.76 × 10^−7^; see Section 4.2) between both highlighted regions in panels (**A**,**B**). Interruptions in hydropathy lines denote gaps in the alignment. Note that these proteins are close homologs (both map to TC system 1.A.1.24.3) and align well throughout their length (SSEARCH E-value: 5.4 × 10^−53^ and coverage: 93%).

**Figure 6 ijms-25-05005-f006:**
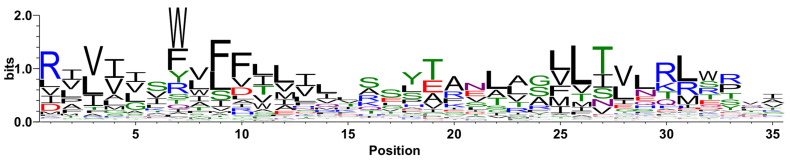
Signature region for families pLIC, VPC and GIC. Sequence logo of the signature region (SR) identified (see Section 4.5). The motif is 35 residues long (E-value: 1.6 × 10^−863^).

**Figure 7 ijms-25-05005-f007:**
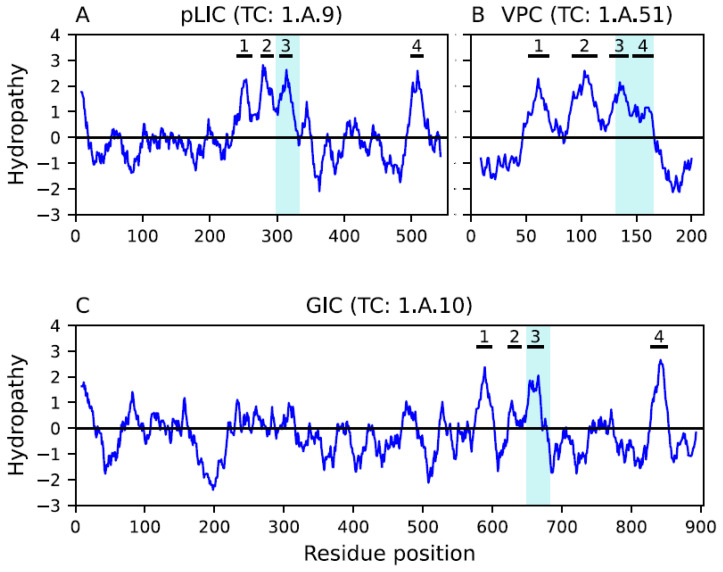
Location of the identified signature region. The signature region (SR; cyan bars) is mapped onto the hydropathy profiles of representative members of each family. Hydrophobic peaks are marked and enumerated: (**A**) pLIC member nicotinic acetylcholine receptor α1 subunit (a1nAChR; QEM43362; TC: 1.A.9.1.21); (**B**) VPC member voltage-gated hydrogen channel 1 (XP_001825565; TC: 1.A.51.1.7); and (**C**) GIC member glutamate ionotropic receptor NMDA subunit 1 (GRIN1; Q91977; TC: 1.A.10.1.12).

**Figure 8 ijms-25-05005-f008:**
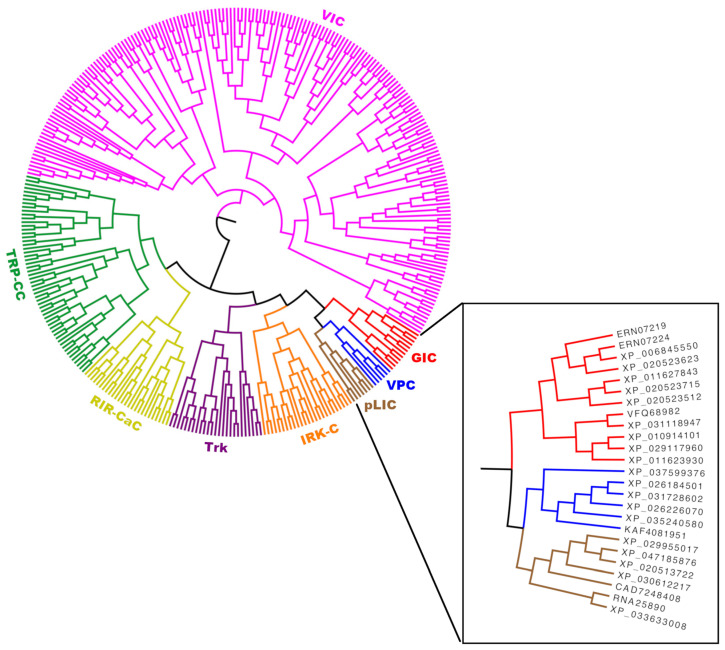
Radial tree of the VIC superfamily. Relationship among all seven families within the VIC superfamily and the pLIC family. Representative homologs for families VPC, GIC and pLIC were selected as described in Section 4.6. For the rest of the families, we used established family members from TCDB. Each family is labeled and shown in different colors. Note that families GIC, VPC and pLIC are sharing the same major branch, which emphasizes their relatedness. Only transmembrane domains were considered to build this tree. The inset identifies the proteins in the pLIC, VPC and GIC clusters. Given that the lengths of the branches are not meaningful, the scale bar was removed.

**Table 1 ijms-25-05005-t001:** The Voltage-gated Ion Channel (VIC) Superfamily. The TCID (column 1), full name (column 2), and abbreviation (column 3) of all families within the VIC superfamily. The last row describes the pLIC family, which is not a currently recognized member of the VIC superfamily, but it is the focus of this study.

TCID	Name	Abbreviation
**The Voltage-gated Ion Channel (VIC) Superfamily**		
1.A.1	The Voltage-gated Ion Channel Superfamily	VIC
1.A.2	The Inward Rectifier K^+^ Channel Family *	IRK-C
1.A.3	The Ryanodine-Inositol 1,4,5-trisphosphate Receptor Ca^2+^ Channel Family	RIR-CaC
1.A.4	The Transient Receptor Potential Ca^2+^ Channel Family	TRP-CC
1.A.10	The Glutamate-gated Ion Channel Family of Neurotransmitter Receptors	GIC
1.A.51	The Voltage-gated Proton Channel Family *	VPC
2.A.38	The K^+^ Transporter Family	Trk
**Candidate new family**		
1.A.9	The Neurotransmitter Receptor Cys loop Ligand-gated Ion Channel Family	pLIC

* TC family 1.A.2 (IRK-C) and TC family 1.A.51 (VPC) do not have the full six TMSs that characterize most members of the VIC superfamily. The latter have both an N-terminal four TMS voltage sensor domain, and a C-terminal two TMS channel domain with a reentrant loop between the two TMSs. Instead, IRK-C proteins have only the channel domain (two TMSs separated by a re-entrance loop), but not the voltage-sensor domain, thus having only two TMSs. In some proteins of the IRK-C family, this domain has been duplicated, so these proteins have four TMSs, each pair being separated by a reentrant loop. By contrast, the TC family 1.A.51 (VPC) includes members that consist only of the first four TMSs of the voltage-sensor domain and, therefore, lack the channel domain of most VIC superfamily protein members. Nevertheless, these VPC family proteins are proton channels, and the voltage-sensing domains of other VIC superfamily proteins can be mutated to gain H^+^ transport function [20,21].

## Data Availability

The original contributions presented in the study are included in the article/Appendix A; further inquiries can be directed to the corresponding authors.

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
