# Peer review of "The Pentameric Ligand-Gated Ion Channel Family: A New Member of the Voltage Gated Ion Channel Superfamily?"

_ijms, 2024, doi:10.3390/ijms25095005_

Round 1

Reviewer 1 Report

Comments and Suggestions for Authors

This is an interesting study that uses bioinformatic data to provide evidence that the Cys-loop ligand-gated ion channel (LIC) family is a member of the voltage-gated ion channel (VIC) family. The authors further suggest that one of the VIC transmembrane segments evolved into a re-entrant loop, which is a fascinating hypothesis. ( Indeed, given this hypothesis, LIC must have evolved first, so is the VIC family a part of the LIC family rather than vice versa?  )

It is mostly well written, although the introduction is overlong and overdetailed, especially for proteins that are barely ( or not at all)  mentioned later. There are also some inaccuracies e.g. lines 144-5:  in  GABA receptors TMS1 does not release ions and substrates or unwind, and indeed the reference there refers to GABA transporters.

 I am surprised by the use of 10ED as a LIC structural comparison ( see data in Table S1 and S2) as this is a low resolution structure published many years ago and has a number of inaccuracies. Is there a reason why this was chosen and would the data be similar if a later structure was used? 

It would also be very useful to the reader if the names of the proteins they used were given in the text as well as their ID ( e.g. line 229 2MAW = a7nAChR)

Specific points

Title : A more accurate wording of ‘The neurotransmitter receptor Cys-loop ligand-gated ion channel family ‘would be ‘the pentameric ligand-gated ion channel family’ and ideally LIC should be pLGIC  ( although pLIC is clearer than LIC as there are LIC that are not part of this family)

L152 aimed not aims

From line 188: more information is needed on signature sequences as a tool to probe homology, both in general and more specifically for VICs e.g.  why is this in the 3rd TMS?

L319 in what regard is the alignment significant– needs clarifying

Fig 5.& Fig 7 There were not any coloured bars in my version

L404 what is ‘the signal of homology’ needs clarifying

Fig 8 It would be useful to know which receptors were at the termini of the tree for LIC, and also for VPC and GIC – I suggest expanding this region as an inset and labelling the receptors

Author Response

Please, see attached file with our answers.

Reviewer 2 Report

Comments and Suggestions for Authors

The authors present evidence that Ligand-gated Ion Channel (LIC) Family is a member of the Voltage-gated Ion Channel (VIC) Superfamily. The authors use a range of homology and alignement techniques to support their claim. However as noted by the authors themselves their hypothesis is not supported by several lines of evidence.

1) They were not able to obtain 3D structural alignments between the LIC and GIC famalies;

2) The phylogenetic analysis of the VIC superfamily and the LIC family did not produce a reliable tree.

Both these observations argue against including the LIC family within the VIC family.

Overall the claims made by the authors are circumstantial at present and are not strong enough to support the conclusion. In particular the evidence around the signature sequence is unconvincing and should not be included. 

The introduction was more like a a "mini-review". It is way too long and unnecessary in its current format. It needs to be shorten considerably. 

Comments on the Quality of English Language

The English in the manuscript is fine.

Author Response

Please, see attached file with our responses to Reviewer 2.
